# The megabase-scale crossover landscape is largely independent of sequence divergence

Qichao Lian [1], Victor Solier[1], Birgit Walkemeier[1], Stéphanie Durand[1], Bruno Huettel [2], Korbinian Schneeberger [1,3 ✉] & Raphael Mercier [1 ✉]

Meiotic recombination frequency varies along chromosomes and strongly correlates with sequence divergence. However, the causal relationship between recombination landscapes and polymorphisms is unclear. Here, we characterize the genome-wide recombination landscape in the quasi-absence of polymorphisms, using *Arabidopsis thaliana* homozygous inbred lines in which a few hundred genetic markers were introduced through mutagenesis. We find that megabase-scale recombination landscapes in inbred lines are strikingly similar to the recombination landscapes in hybrids, with the notable exception of heterozygous large rearrangements where recombination is prevented locally. In addition, the megabase-scale recombination landscape can be largely explained by chromatin features. Our results show that polymorphisms are not a major determinant of the shape of the megabase-scale recombination landscape but rather favour alternative models in which recombination and chromatin shape sequence divergence across the genome.

[1] Department of Chromosome Biology, Max Planck Institute for Plant Breeding Research, Carl-von-Linné-Weg 10, 50829 Cologne, Germany. [2] Max Planck-Genome-centre Cologne, Max Planck Institute for Plant Breeding Research, Carl-von-Linné-Weg 10, 50829 Cologne, Germany. [3] Faculty of Biology, LMU Munich, 82152 Planegg-Martinsried, Germany. ✉email: schneeberger@mpipz.mpg.de; mercier@mpipz.mpg.de

**M**eiotic recombination is initiated by the formation of numerous DNA double-strand breaks, a minority of which are repaired as crossovers (COs), resulting in reshuffling of the genetic material between generations. COs are, thus, crucial for diversity, adaptation, evolution and breeding[1–4]. Two pathways have been described for meiotic CO formation (class I and II)[1,3,4]. Class I COs represent the vast majority of COs and are subject to interference, the propensity of COs to be widely spaced along chromosomes[5].

COs are not homogeneously distributed and recombination frequencies vary along chromosomes[6,7]. Many different features are correlated with the recombination landscape. One consistent pattern across monocentric species is the suppression of COs at and next to centromeres[3,8,9]. The landscape can also differ between the two sexes of the same species, a phenomenon called heterochiasmy[10–13]. Polymorphism between homologues can negatively affect crossovers, as observed very locally at crossover hotspots or even completely suppress crossovers in cases of large polymorphisms, like megabase-scale inversions[7,14–21]. In contrast, however, heterozygous regions in *Arabidopsis thaliana* showed increased recombination rates when juxtaposed with homozygous regions, suggesting that the density of small-scale sequence divergence can increase recombination rates[22]. In addition, increasing single nucleotide polymorphism (SNP) density in hybrids associates positively with COs, and the pericentromeric regions that are dense in polymorphisms are also elevated in COs, potentially due to a positive feedback of mismatch recognition during CO formation[21]. A positive correlation between polymorphisms and recombination landscapes can also be observed in natural populations: in many species, historical recombination landscapes as deduced from linkage disequilibrium are positively correlated with SNP densities[23–27]. In addition, COs tend to colocalize with gene promoters and with regions of open chromatin and low levels of DNA methylation[3,28–30].

To better understand the relationship between polymorphisms and meiotic recombination, we aimed to compare CO distribution along chromosomes in the quasi-absence (inbred lines) and presence (hybrids) of polymorphisms. In hybrids, the numerous DNA polymorphisms can be used to precisely map COs[6,30–39], while this is not an option in homozygous inbred lines. Instead, CO frequency in such lines can be estimated by cytological techniques[40–43], but this has also some limitations, such as the difficulty in identifying individual chromosomes. Alternatively, fluorescence-tagged lines (FTLs) could be used to measure recombination in intervals flanked by markers conferring fluorescence in seeds or pollen grains, but these FTLs are not suitable for mapping the genome-wide CO landscape[44–46].

In this study, we develop a method to analyse genome-wide recombination landscapes in inbred lines. We characterize the crossover landscapes of two Arabidopsis inbred lines and compare them to the hybrid, and with the historical recombination pattern in this species. All these CO landscapes are remarkably similar, with the exception of local suppression due to large heterozygous rearrangements. This shows that polymorphism density, with the exception of large structural variations, is not a major determinant of the CO landscape. We also show that only very few chromatin features, like chromatin accessibility and DNA methylation, are sufficient to explain more than 85% of the megabase-scale recombination landscape in Arabidopsis.

## Results

**A method to robustly detect crossover genome-wide in pure lines**. To investigate the landscape of meiotic recombination in *A. thaliana* inbred lines, we applied moderate EMS mutagenesis to introduce genetic markers into the genomes of *A. thaliana* Col-0 and L*er*. Independent M2 mutants were crossed to generate F1*s, and independent F1*s were reciprocally crossed to generate F1 populations, which were used to analyse recombination independently in female and male meiosis. (Fig. 1, Supplementary Fig. 1, Supplementary Table S1, Materials and methods). Through Illumina short-read genome sequencing of F1*s and F1s, we identified 838–955 high-confidence mutations segregating in the Col populations and 471–539 in the L*er* populations (Supplementary Table S1), which is a negligible level compared to the natural divergence between these two accessions[14,39,47]. The markers were randomly distributed across the chromosomes, which allowed the identification of meiotic CO events (Supplementary Fig. 2A–E). We analyzed four independent pairs of populations from both accessions, with a total of 309 and 309 progenies derived from female and male meiosis in Col and 253 and 251 progenies derived from female and male meiosis in L*er*, respectively. Overall, we identified 3155 COs in Col (examples shown in Supplementary Figs. 3 and 4, median resolution 522 kb, Supplementary Data 2) and 2004 (median resolution 855 kb, Supplementary Data 3) COs in L*er*. We observed a consistent CO frequency between the replicate populations (Fig. 2A, B), arguing against the unlikely possibility that an EMS mutation dominantly affected CO numbers in the F1*. CO numbers were correlated neither with sequence depth nor with the number of markers, suggesting an absence of bias in CO detection (Supplementary Fig. 5). Altogether, this suggests that our method robustly detects COs in inbred lines.

**A female and male crossover landscape in the Columbia-Landsberg F1 hybrid.** To compare the recombination landscapes of Col and L*er* with the corresponding F1 hybrid, we sequenced reciprocal back-crosses of Col/L*er* F1 hybrids with Col to identify COs in 428 and 294 progenies derived from female and male meiosis, respectively. We identified 1192 COs (median resolution

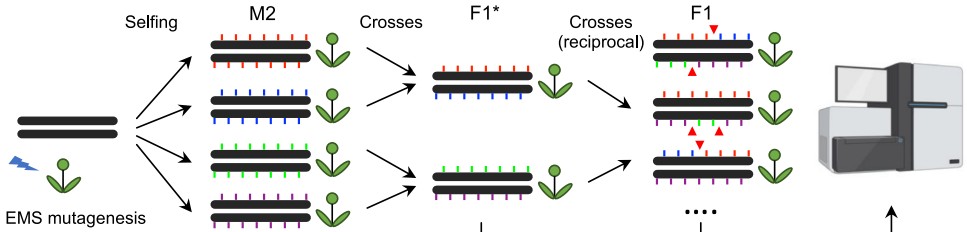

**Fig. 1 Experimental design for CO identification in Arabidopsis inbred lines.** M2 plants were derived from the selfing of independent EMS-treated seeds. Pairs of M2s were crossed to produce F1*s, which are then heterozygous for a set of unique mutations defining two phases, indicated by coloured ticks. Two F1*s were then reciprocally crossed to generate F1 populations. The DNA of leaves of F1* and F1 plants were sequenced using Illumina. The colour-coded ticks indicate EMS-induced mutation markers. The red triangles represent COs, detected by phase switching. This design allows the detection of COs that occurred in the female and male meiocytes of the F1* plants. The sequencer figure was created with BioRender.com.

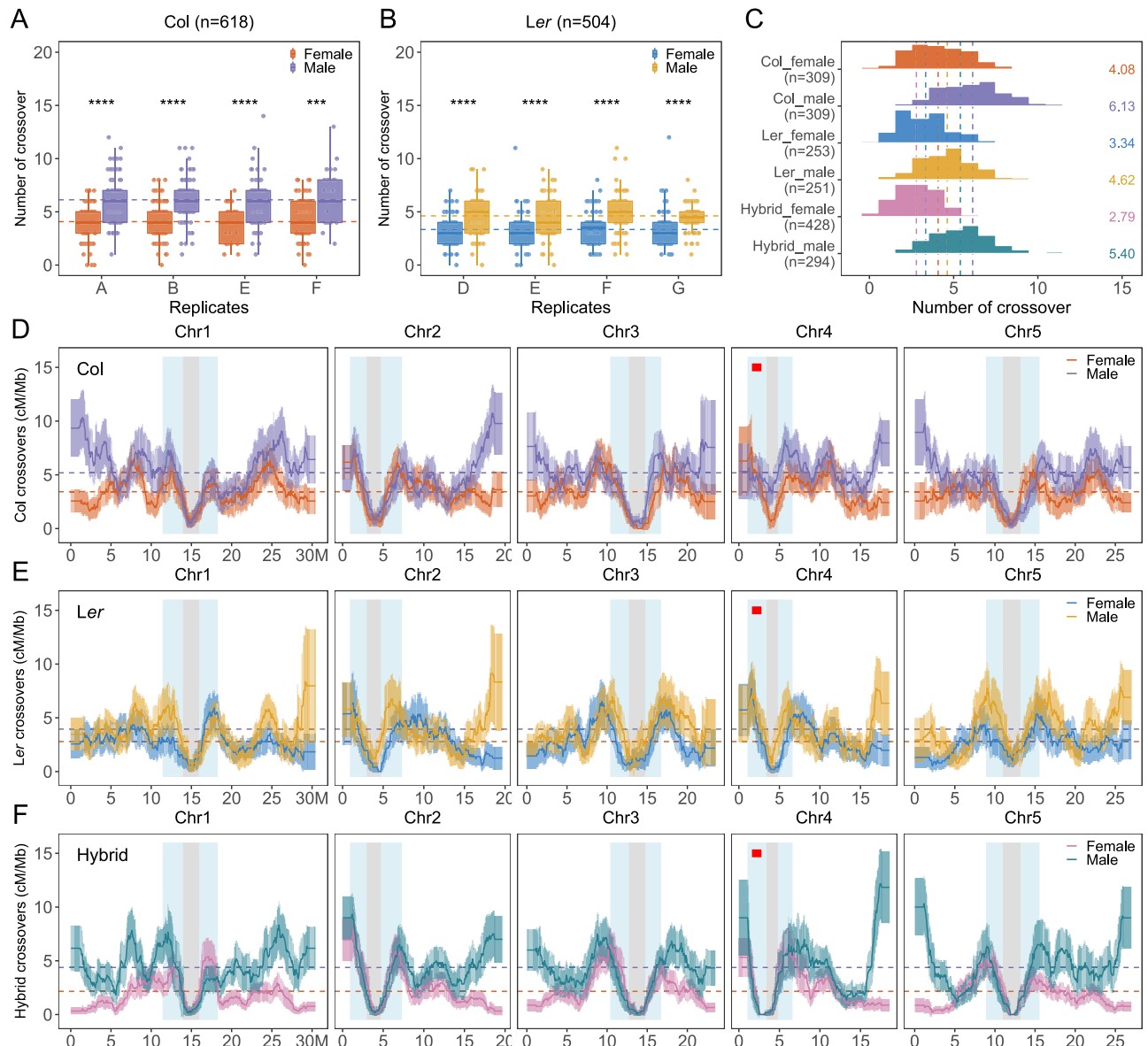

**Fig. 2 Analysis of female and male COs in Col, Ler, and F1 hybrid populations. A**, **B** The numbers of COs per gamete in each replicate population of Col and Ler, respectively. Each letter corresponds to one F1* plant (Fig. 1). The two-sided Mann–Whitney test was used to evaluate the differences in CO numbers between female and male meiosis. The p values for Col replicate **A**, **B**, **E**, **F** are 4.9e−12, 2.5e−12, 4.7e−06 and 1.9e−04, respectively. The p values for Ler replicate **D**–**G** are 5.6e−06, 6.7e−05, 6.8e−05 and 2.6e−05, respectively. In the boxplot, the centre line indicates median, the bounds of box indicate 25th and 75th percentiles, the whiskers indicate 1.5 * IQR (IQR: the distance between the first and third quartiles). **C** Distribution of CO number per gamete in female and male meiosis of Col, Ler and F1 hybrids. The mean number of COs are colour-coded and indicated by dashed lines. The sample sizes are indicated in parentheses. **D**–**F** The chromosomal distribution (sliding window-based, window size 2 Mb, step size 50 kb) of COs in female and male meiosis of Col, Ler, and F1 hybrids, with 90% confidence interval. The genome-wide mean level of recombination is shown with dashed lines. The pericentromeric and centromeric regions are indicated by grey and blue shading, separately. The ~1.2 Mb inversion between Col and Ler on chromosome 4 is indicated by a red bar. Source Data are provided as a Source Data file.

739 bp) and 1587 COs (median resolution 1019 bp) in female and male hybrids, respectively (Supplementary Fig. 6, Supplementary Data 4). The female and male high-resolution CO distribution that we obtained is consistent with a previous dataset that described female/male CO landscapes with lower resolution[11] and CO distribution in the same hybrid in F2s that does not distinguish female and male COs[7] (Supplementary Fig. 7 and 8, Supplementary Data 5). Comparison of the genomic compartments where COs occurred did not reveal differences between females and males, with COs notably enriched in promotor regions in

both sexes. This suggests that the factors driving fine-scale CO placement are similar in female and male meiosis (Supplementary Fig. 7E, F).

**Comparing crossover number in *Arabidopsis* pure lines and hybrids.** In all three types of populations, Col, Ler and hybrid, we observed heterochiasmy, i.e., significantly more COs in male compared to female meiosis (Mann-Whitney test, $p < 2.2e−16$, Fig. 2C). This heterochiasmy was confirmed in the three

genotypes by counting MLH1 foci, whose number is consistently higher in male compared to female meiocytes (Supplementary Fig. 9). In male meiosis, both the highest (Col, 6.13) and the lowest (Ler, 4.62) numbers of COs are observed in inbred lines, with the hybrid exhibiting an intermediate number of COs (5.4), consistent with MLH1 foci analysis (Fig. 2C, Supplementary Fig. 9). The observation that the hybrid has an intermediate number of COs compared to the two inbred lines suggests that the global CO frequency in males is mainly genetically controlled in trans and not, or only marginally, driven by sequence polymorphism. In females, the highest CO number is also observed in Col (4.08), with less COs in Ler (3.34, $p = 1.2e-07$), indicating that the same trans mechanism also influences CO frequency in females. However, an even lower level of COs is observed in the hybrid (2.79, $p = 0.0002$), suggesting that an additional phenomenon is responsible for the reduced CO frequency specifically in female hybrids. In all contexts, CO interference is more pronounced in females than in males, with the strongest interference observed in female hybrids (Supplementary Figs. 10–12). It should be noted that the CO interference was measured within DNA (Mb) space and that chromosomes are organized along shorter axes in females than in males[48]. When a conversion is applied for analysis in the chromosome axis space (μm)[49], CO interference appears very similar in female and male meiosis (Supplementary Fig. 12), suggesting that interference propagates at similar μm distances along axes in females and males but that due to higher compaction interference acts over larger DNA distances in females. The anti-correlation observed between CO interference and CO numbers suggests that modulation of CO interference, likely through modulation of axis length, is an important determinant of CO numbers. In both sexes of the three backgrounds, CO number is positively correlated with individual chromosome length, except for the female hybrid where the curve is almost flat at just above 0.5 COs per chromosome per gamete, corresponding to one CO per bivalent and a very strong CO interference (Supplementary Fig. 13).

**Polymorphisms does not define megabase-scale crossover distribution.** Along chromosomes, a strikingly similar pattern is observed in the three genetic backgrounds. COs are markedly suppressed at the centromeric regions and tend to be frequent at the edge of peri-centromeres in both female and male meiosis. In all three backgrounds, the female and male recombination landscapes tend to diverge with decreasing distance from telomeres, with distal regions exhibiting among the highest recombination intervals in males and the lowest in females (Fig. 2D–F, Supplementary Fig. 14). The female/male difference is less pronounced in Ler, notably in distal regions. This may be due to a generally lower frequency of COs in Ler compared to Col, and because trans-factors (e.g., *HEI10*) tend to affect more distal regions[50]. However, it should be noted that the Ler profile tends to have a larger interval of confidence, notably at telomeres, because of slightly smaller sample size and marker set than the two other genotypes. Strikingly, CO distributions are more closely correlated between the same sexes across the three different backgrounds than between the two sexes in the same background (Fig. 2, Supplementary Fig. 16). For example, female hybrids are more similar to female Col and Ler (Spearman's correlation $r_s = 0.62$ and $0.64$) than to male hybrids ($r_s = 0.26$). Thus, sequence divergence appears to have a far lesser impact on the CO landscape than the sex of the meiocyte.

To compare the contemporary CO landscapes with the historical landscape, we reconstructed a historical recombination map using a set of non-singleton SNPs generated from 2029 accessions (Fig. 3A and Supplementary Fig. 15)[51,52]. Confirming previous findings[27], the historical CO landscape is strongly correlated with the sequence diversity (Fig. 3A, F, Supplementary Fig. 16). The historical landscape is the result of combined female and male recombination, and we thus compared it to the merged female and male dataset for the inbred lines and to the previously described large Col/Ler F2 dataset (Fig. 3D–F and Supplementary Fig. 16)[7]. To facilitate the comparison of the landscapes independently of total CO numbers, we show both the observed CO density (cM/Mb, Fig. 3B) and the corresponding normalized distribution (Fig. 3A, C). Strikingly, the CO landscape in the two inbred lines and hybrid all appear similar to each other, with co-localization of many peaks and valleys, including large peaks on both sides of the centromeric regions, but also in the middle of the arms (Fig. 3C). This similarity is confirmed by genome-wide correlation analyses (Fig. 3D–F). Correlation between CO levels in intervals in Col vs Ler is 0.66 (Fig. 3E), and 0.74 between Col and the hybrid (Fig. 3D). The coefficient of correlation is even higher if non-linear correlation is used ($r_n = 0.73$–$0.78$, Fig. 3F) or when chromosome arms and peri-centromeres are considered separately (Fig. 4A, Supplementary Fig. 18). The historical recombination landscape is also similar to the three contemporary landscapes, with most peaks being conserved ($r_n = 0.6$–$0.7$, Fig. 3A, F). This shows that the global CO landscape is largely independent of the presence (hybrid and historical) or quasi-absence (Col and Ler) of polymorphisms between the two chromosomes that recombined.

While the CO landscapes are similar, they are not identical. One notable divergence is observed at position ~2 Mb of chromosome 4, with suppression of CO in the hybrid that is not observed in the inbred lines and the historical landscape (Figs. 2D–F and 3A–E). Accordingly, the corresponding intervals stand out in the correlation analysis (red arrow, Fig. 3D). This region corresponds to a large ~1.2 Mb genomic inversion, which suppresses recombination in the Col/Ler hybrid where it is heterozygous[7,47,53,54]. We then asked if the smaller rearranged regions are also depleted for COs. We explored the overlap of COs with the non-syntenic regions by employing permutation tests in F2 hybrids and observed a strong depletion of COs in non-syntenic regions (Supplementary Fig. 17B, $p = 0.0002$) and CO enrichment in the adjacent regions (Supplementary Fig. 17C, $p = 0.0002$), confirming that structural arrangements are correlated with inhibition of CO formation in hybrids[7,14]. The CO resolution obtained for inbred lines did not allow us to test if these regions recombine normally in Col and Ler, as is the case for the unique large inversion.

Consistent with previous analyses[27], we found that the historical recombination rate is highly correlated with sequence diversity along chromosomes (Fig. 3A, F, Supplementary Fig. 16), and contemporary COs in the Col/Ler hybrid are correlated with SNP density between Col and Ler[28,55]. As shown above, the CO landscapes all show high similarity to each other. Consequently, the CO landscapes in Col and Ler are correlated with Col/Ler SNPs ($r_n = 0.28$–$0.39$) and sequence diversity ($r_n = 0.44$–$0.45$), whereas these polymorphisms were absent in the Col and Ler inbred lines where these COs were produced. This strongly argues against the possibility that the polymorphisms shape the CO landscape, as the CO landscape is largely unchanged when polymorphisms are absent (with the notable exception of large rearrangements).

**Association of crossover landscape with genomic and epigenomic features.** To decipher the contributions of genomic and epigenomic features to shaping the CO landscape, we analyzed the recombination distribution in Col with a total of 17 different features. The CO and the genomic and the epigenomic data were

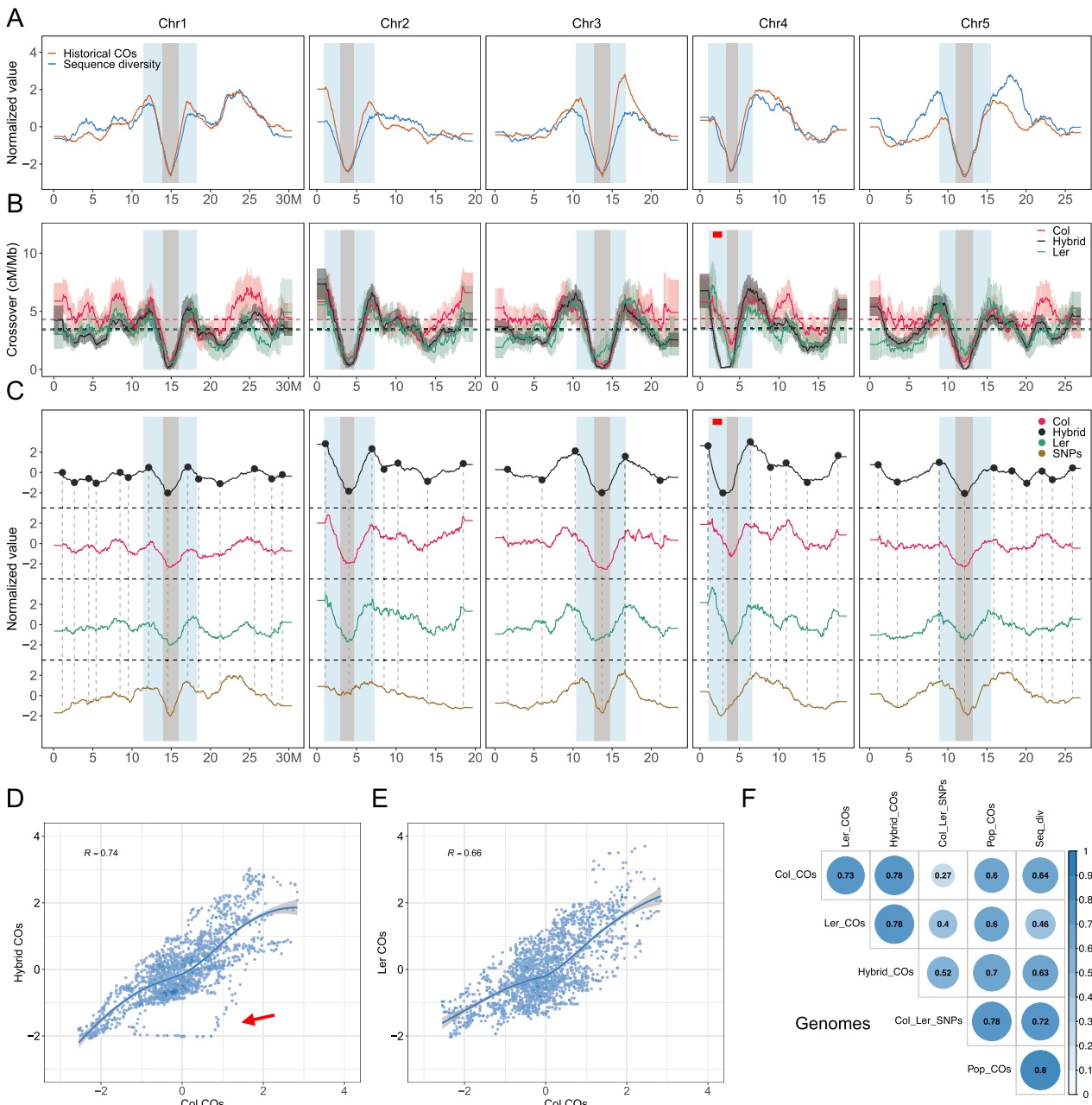

**Fig. 3 Comparison of the genome-wide CO landscape in Col, Ler, and F2 hybrids with genetic polymorphisms. A** The normalized landscapes (sliding window-based, window size 2 Mb and step size 50 kb) of historical recombination rate (4Ner per kb, red) and sequence diversity (π, blue) within 2029 Arabidopsis accessions across genomes. **B** The chromosomal distribution of COs, with 90% confidence interval in Col, Ler (merged female and male), and Col/Ler F2 hybrids. The genome-wide mean-level recombination is shown with dashed lines. **C** The z-score-normalized distribution of COs in Col, Ler, and F2 hybrids and the density of SNPs between Col and Ler along chromosomes. The relative crossover frequency (the number of crossovers in the given window divided by the total crossover number within the respective chromosome) was calculated for each individual chromosome. The peaks and valleys detected in the hybrid landscape are represented by black points prolonged by vertical dashed lines. The pericentromeric and centromeric regions are indicated by grey and blue shading, separately. The ~1.2 Mb inversion between Col and Ler on chromosome 4 is represented by a red bar. **D, E** Spearman's correlation analysis of the normalized distribution of COs in Col, Ler and Col/Ler F2 hybrids. The red arrow points to the effect of the ~1.2 Mb inversion between Col and Ler on chromosome 4, which is only present in the comparison between inbred lines and hybrids. **F** The non-linear correlation coefficient matrices among genome-wide distributions. Col_COs, Ler_COs, Hybrid_Cos and pop_COs represent CO landscapes in Col, Ler, F2 hybrids and the population of 2029 Arabidopsis accessions (historical recombination rate); SNPs represents SNP density between Col and Ler, and Seq_div represents sequence diversity in the population of 2029 Arabidopsis accessions. Source Data are provided as a Source Data file.

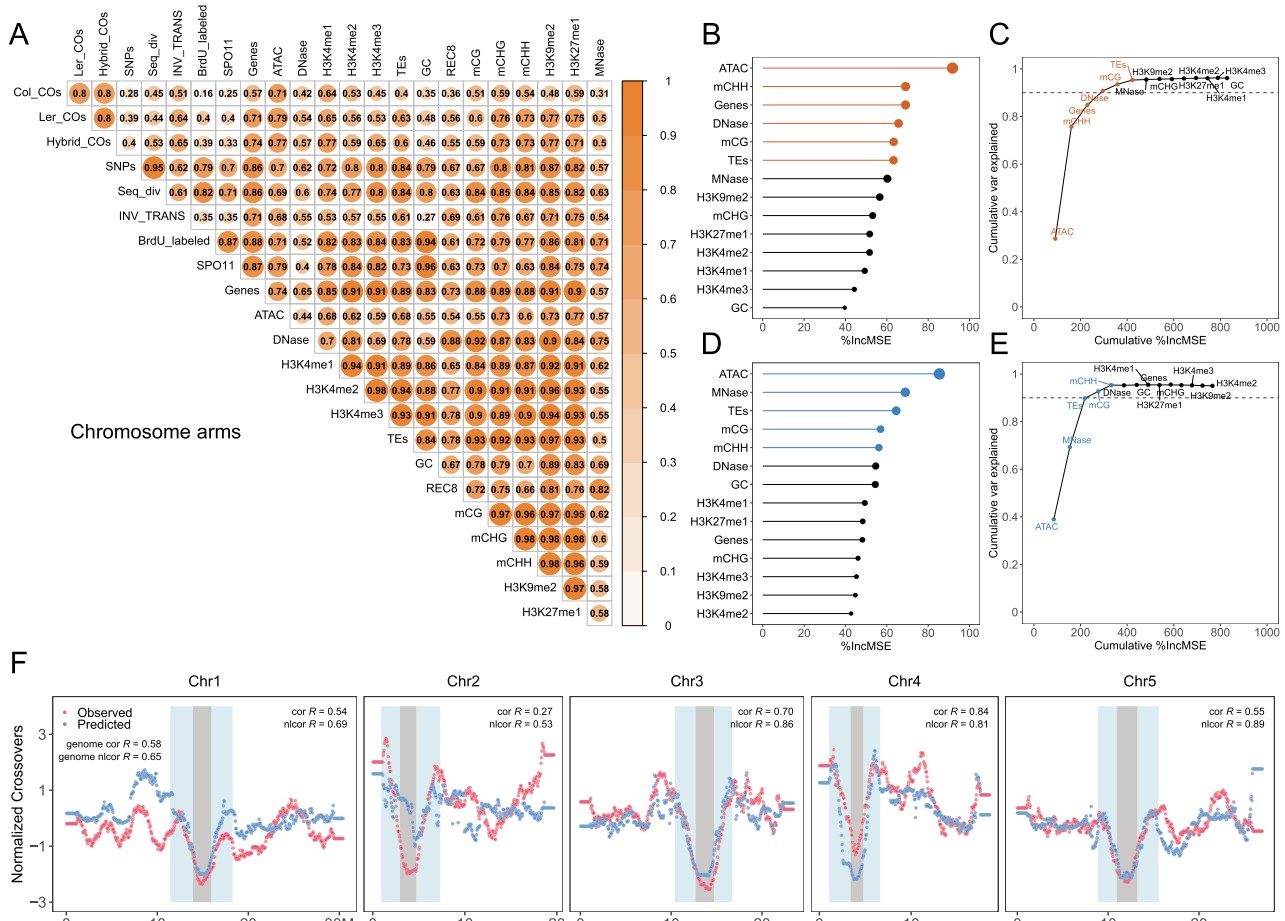

**Fig. 4 Association and prediction of CO distribution with genomic and epigenomic features. A** The non-linear correlation matrices show the comparison of pairwise features along chromosome arms, with differences in colour and size according to the correlation scale. Col_COs, L*er*_COs and Hybrid_COs (CO landscapes in Col, L*er*, and F2 hybrids), SNPs (SNPs density between Col and L*er*), Seq_div (sequence diversity in the population of 2029 Arabidopsis accessions), INV_TRANS (inversions and translocations between Col and L*er*), BrdU_labelled (origins of DNA replication, $\log_2$(BrdU/gDNA)), SPO11 (SPO11-1-oligos, $\log_2$(oligos/gDNA)), Genes, TEs and GC (gene, TE and GC density), ATAC and DNase (chromatin accessibility, ATAC-seq and DNase-seq, $\log_2$(Tn5/gDNA) and $\log_2$(DNase/gDNA)), H3K4me1/2/3, H3K9me2, H3K27me1 (euchromatin, heterochromatin, and Polycomb histone marks, ChIP-seq, $\log_2$(ChIP/input)), REC8 (cohesin, ChIP-seq, $\log_2$(ChIP/input)), mCG, mCHG and mCHH (DNA methylation in CG, CHG, and CHH contexts, proportion methylated cytosine), MNase (nucleosome occupancy, MNase-seq, $\log_2$(MNase/gDNA)). The importance of each of the 14 features for explaining variation in CO distribution at the chromosome-arm (**B**) and genome scale (**D**), respectively. The size of points corresponds to the importance. The cumulated proportion of variation that can be explained with the features at the chromosome-arm (**C**) and genome scale (**E**), respectively. The top six and five most important features, for which the cumulative proportion of variation that can be explained reaches the plateau, are coloured separately. **F** The chromosomal distribution of observed and predicted COs. The CO profiles of individual chromosomes were predicted using profiles of the top five most important features from the other four chromosomes. The Spearman's and non-linear correlation coefficients between the predicted and observed CO distributions for each chromosome and the whole genome are indicated, respectively. Source Data are provided as a Source Data file.

all produced in the same strain, Col. The measured features included GC content; gene and transposable element densities; origins of DNA replication (BrdU-seq)[56]; meiotic DSBs (SPO11-1-oligonucleotides)[57]; chromatin accessibility in flowers (ATAC-seq and DNase-seq)[58–61]; euchromatic (H3K4me1, H3K4me2 and H3K4me3, ChIP-seq)[57,62] and heterochromatin histone modification marks in flower buds (H3K9me2 and H3K27me1, ChIP-seq)[62]; DNA methylation in male meiocytes (CG, CHG and CHH contexts, BS-seq)[63]; nucleosome occupancy in buds (micrococcal nuclease sequencing, MNase-seq)[57]; and the meiotic cohesin REC8 occupancy (ChIP-seq, Fig. 4A, Supplementary Figs. 18–20, Supplementary Data 1)[62].

Genome-wide, CO distribution is correlated with many genetic and epigenetic features, notably positively with open chromatin (ATAC, $r_n = 0.71$), H3K4me1($r_n = 0.65$), gene density ($r_n = 0.64$) and CHG methylation ($r_n = 0.55$, Supplementary Figs. 18–20).

These correlations are at least partially driven by the centromeric regions, at which COs are abolished and where these features are strongly depleted (Supplementary Figs. 18–20). However, considering only the chromosome arms, the correlations are almost the same between COs and ATAC ($r_n = 0.71$), H3K4me1($r_n = 0.64$), CHG methylation ($r_n = 0.59$), gene density ($r_n = 0.57$) and other features (Fig. 4A), suggesting a relationship between CO density and chromatin features beyond the centromere.

We next used a machine-learning algorithm (random forest) to analyse the contribution of the 14 chromatin features to explaining the variation in the crossover landscape in Col. We first developed a model to predict the frequency of meiotic recombination for a given interval with all the chromatin features together and analyzed how the model learned to perform the prediction. Over 95% of the variation can be explained by the random forest predictive model (Fig. 4C, E). As shown in Fig. 4B, D, the most important feature was

open chromatin (ATAC), which alone explained 39% of the genome-wide variation (Fig. 4E) and 29% of the variation along chromosome arms (Fig. 4C). We observed that the top three features can explain >85% of the variation, while the top six features and five features can explain ~95% of the variation along chromosome arms and genome-wide, respectively (Fig. 4C, E). In order to further investigate the performance of the random forest model, we used four chromosomes as the training set and the remaining chromosome as the testing set. This analysis was done for each of the five chromosomes, considering the top five features for the entire genome (Fig. 4F) or the top six features for chromosome arms (Supplementary Fig. 21). The model trained with the training set performed well with the test set, resulting in a significant correlation ($r_s = 0.58$, $r_n = 0.65$) between the prediction and the observations of the test set (Fig. 4F, Supplementary Fig. 21). Altogether, these results show that it is sufficient to use only a few chromatin-related features including chromatin accessibility and DNA methylation, to predict a large part of the megabase-scale distribution of meiotic recombination in *A. thaliana*.

## Discussion

In this work, we developed a method to analyse the genome-wide recombination landscape in inbred lines and applied it to the Arabidopsis accessions Col and L*er*. This method is based on the introduction of a limited number of markers and allows robust detection of COs. The strategy can be applied to any species for which homozygous lines and mutagenesis are available. We expect this method to be particularly useful for exploring the natural variation of recombination landscapes in species that are inbred lines in the wild (e.g., Arabidopsis) and for exploring CO distribution in species where inter-strain crosses are problematic (e.g., in the fission yeast *Schizosaccharomyces pombe* because of killer meiotic drivers)[64].

Meiotic recombination frequency has previously been studied in hybrids in many species and varies along chromosomes and positively correlates with the distribution of polymorphisms[3,7,12,21,23–27,55]. One of the possible causes for these correlations is that heterozygosity may favour the formation of COs, in a process putatively driven by mismatch recognition during DSB repair[21,22]. In fact, mutants without mismatch sensor function showed a reshuffling of meiotic recombination towards regions with less polymorphisms[21], which suggests that polymorphisms are involved in the local placement of COs. However, the broad distribution of COs across the chromosomes was only marginally affected in mismatch recognition mutants, which is in agreement with chromatin being the major determinant of the megabase-scale recombination landscape.

We showed here that the megabase-scale recombination landscape in inbred lines is similar to those of hybrids as well as to historical patterns. Broad conservation of CO distribution was previously suggested in tomato and maize by comparing recombination nodules in inbreds to genetic maps in hybrids[41,42,65]. The observation that the CO landscape is maintained in the quasi-absence of polymorphism leads to the conclusion that polymorphisms are not a major determinant of the megabase-scale CO distribution. Polymorphisms, including SNPs and small rearrangements, influence the local recombination pattern[18–21], but this effect is not manifest at the megabase-scale; at this range, the landscape appears to be largely unaffected by polymorphism density. An important exception is genomic rearrangements, such as the ~1.2 Mb inversion (between Col and L*er*) on chromosome 4, where COs are abolished in hybrids, while the corresponding regions are CO-proficient in isogenic lines. Smaller structural variations are also associated with CO depletion in the hybrid and are presumably CO-prone in the inbred lines, though we cannot confirm this because of the relatively low resolution in CO position.

In many species, COs tend to colocalize with nucleosome- and methylation-depleted gene promoters[3,7,28,30,36,66,67], consistent with our observation in Col/L*er* hybrids. Moreover, in this study, we found that among a total of 14 genomic and epigenomic features, open chromatin (ATAC), DNA methylation in the CHH context, and gene density, are the most potent factors for predicting the distribution of COs along chromosomes arms in inbred Col, which is consistent with previous findings[57,62]. Interestingly, these three features were enough to explain ~85% of the variation of the CO distribution along chromosome arms. We do not claim that these three features alone directly control CO positions. For example, if gene density is ignored, the top three features can still explain more than 85% of the variation (ATAC, mCHH and MNase, Supplementary Fig. 22). Our results show that the chromatin context, which can be largely captured using only a few features, can robustly predict megabase CO landscapes. Interestingly, the most predictive feature (open chromatin, ATAC), is largely conserved between different tissues at the megabase-scale (Supplementary Fig. 23). This suggests that the megabase-scale chromatin landscape is stable throughout development and is a major driver of the CO landscape.

Our results suggest that the large-scale CO landscape is not driven by the polymorphism density. Thus, two possibilities may explain the correlation between polymorphism density and recombination observed in hybrid and historical landscapes. First, the recombination landscape could gradually shape the polymorphism density. Indeed, meiotic recombination is mutagenic, which might be an important driver of genetic diversity and genome evolution[25,31,67–71]. In addition, selection tends to reduce polymorphisms in regions with low recombination rates: both the spread of beneficial mutations and the removal of deleterious mutations by selection reduce polymorphism levels and this effect is larger if recombination is low[72]. A second, not mutually exclusive hypothesis, is that local differences in chromatin features not only influence the distribution of recombination, but that chromatin, independently of recombination, contributes to genomic diversity by shaping differences in local mutation rates along the genome[73,74].

While the recombination landscape is largely conserved between the inbred lines and the hybrid, they differ in the total CO number. Globally, there are more COs in Col than in L*er*, with the hybrid having an intermediate number. This is consistent with previous observations in a few crossover reporter intervals and is probably largely driven by an allelic difference in the pro-CO factor HEI10; the Col allele was shown to increase the number of COs compared to the L*er* allele in a co-dominant manner[50]. Other trans components, such as the SMC5/6 complex subunit SIN1, probably also contribute to the difference in recombination between Col and L*er*[75].

When female and male recombination are analyzed separately, CO rates and MLH1 foci are always highest in Col and lowest in L*er* and always higher in males than in females. The male hybrid exhibits an intermediate recombination rate of CO formation, but, in contrast, the female hybrid has less COs than the two inbred lines. This suggests that some mechanism specifically reduces CO frequency in female hybrids compared to the female inbred lines. One possibility is that class II COs, which represent a minority of COs, are inhibited in the presence of polymorphism and thus reduced in hybrids[53,76]. This would have a proportionally larger effect on female meiosis where class I COs are less numerous than in males, and thus account for the very low level of COs in female hybrids. As class II COs are non-interfering, this would also explain why CO interference is stronger in female hybrids than in female inbreds and could especially account for the absence of very closely spaced double-COs (Supplementary Fig. 10)[49]. Interestingly, in female hybrids, the

number of COs observed was close to the obligate one crossover per bivalent (0.5 crossovers per chromatid), suggesting that the CO landscape in female hybrids corresponds to the distribution of the obligate CO, which thus occurs highly preferentially in the proximal regions. The most striking contrast between females and males was the pronounced difference in the distal regions, where males tend to recombine more than females in both pure lines and hybrids. This further confirms that the megabase-scale recombination landscape is largely independent of polymorphisms and instead suggests that the cellular environment plays a much more critical role, notably by controlling chromosome organization[49,77].

An improved understanding of the control of meiotic recombination along the chromosome opens the possibility of manipulating COs and increasing recombination rates globally[53,76,78] and in reluctant regions. This would facilitate the reshuffling of genomic material, breaking of the linkage between beneficial and deleterious alleles and allow the combination of favourable alleles in elite varieties.

## Methods

**Isogenic population construction and sequencing**. Plants were grown in greenhouses or growth chambers (16-h day/8-h night, 20 °C). Wild-type Col-0 and Ler-1 are 186AV1B4 and 213AV1B1 from the Versailles *A. thaliana* stock center (http://publiclines.versailles.inra.fr/). For each accession, seeds were subjected to EMS mutagenesis as described in ref.[79], and four independent M2 plants were crossed to produce two independent F1*s, which were consequently heterozygous for a set of EMS mutations (Fig. 1). Then, the two F1* plants were reciprocally crossed to generate two F1 populations. To test the robustness of the results and detect the unlikely possibility that a dominant modifier of recombination was caused by an EMS-induced mutation, two independent replicates of the entire process were performed for each accession. These F1*s and F1 plants were then used for CO analysis by whole-genome sequencing (Fig. 1, Supplementary Table S1). Leaf samples from the populations were used for DNA purification and library preparation for 2 × 150bp HiSeq 3000 Illumina sequencing[80]. To detect the markers, we sequenced genomic DNA from the F1*s (~59× and ~16×, in Col and Ler, respectively) and F1s (~4.8× and ~5.0×)

**Identifying and genotyping EMS-induced mutations**. For each individually sequenced F1* and F1 plant of *Arabidopsis* Col and Ler accessions, the whole-genome resequencing reads were aligned against the Col-0 TAIR10 reference genome[81,82] and Ler assembled genome[47] by BWA v0.7.15-r1140[83] with default parameters, and variant calling in F1* populations was performed using inGAP-family[39], separately. To obtain high-quality mutation marker lists, we first removed non-allelic markers using inGAP-family with input from the tandem replicates and structural variants predicted using Tandem Repeats Finder v4.09[84] and inGAP-family, respectively, and further filtered variations that did not meet the following criteria: (i) heterozygous genotype with alternative allele frequency from 0.4 to 0.6, (ii) specific to each of the F1*s, and (iii) GA to CT substitution. Then, the read count and genotype map of mutation markers of each F1* was generated from their F1 progenies by inGAP-family, which was subsequently used for mutation phasing and CO identification. In order to properly compare CO landscapes in isogenic and hybrid lines, we transferred the coordinates of mutations in the Ler population to Col-0 by using syntenic alignments identified by SyRI v1.2[85].

Two additional replicates in Col (C and D) were discarded, because the marker analysis showed that one of the F1*s resulted from an accidental selfing and not from a cross. Two additional replicates in Ler were also discarded (C and H), because the number of detected mutation markers (<350) was insufficient for good genome coverage.

**Phasing mutations and CO identification in inbred lines**. To phase the EMS-induced mutation markers, we employed a hierarchical clustering-based sliding window method, with a window size of ten mutation markers and step size of one mutation marker (Supplementary Fig. 1). For each window, the genotype map of the mutation markers was constructed and used as input for clustering, resulting in two groups: one consisting of wild-type samples and one comprising mutant samples. The genotype and phase of mutation was evaluated by the voting strategy based on multiple window clustering. During this process, for the first and last 5-9th markers, a support rate of 0.9 was used to impute and correct the genotype of the marker if it was not covered or poorly covered, and for the other non-covered or poorly covered markers in between, a support rate of 0.8 was used. The CO events were defined as consistent switches of phase of mutation markers along chromosome arms, and the border was further refined by examining the wild-type allele of the mutation. For the termini of chromosomes, COs were validated as switches with one well-supported mutant allele or more than ten reads supported

by the consecutive wild-type allele of the variant marker. The CO interference was analyzed using MADpattern v.1.1[86,87].

**CO analysis in hybrid population**. The Col/Ler and wild-type Col plants were reciprocally crossed to construct female and male populations (428 and 294 plants, respectively). Leaf samples of the backcross populations were collected for DNA purification, library preparation and Illumina sequencing[80]. In addition, the raw reads of the Col/Ler F2 population were downloaded from ArrayExpress with the accession number E-MTAB-8165 (https://www.ebi.ac.uk/arrayexpress/experiments/E-MTAB-8165)[7].

The quality of the raw sequencing datasets was checked using FastQC v0.11.9 (http://www.bioinformatics.babraham.ac.uk/projects/fastqc/), and then adaptors and low-quality bases were trimmed using Trimmomatic v0.38[88] with parameters "LEADING:5 TRAILING:5 SLIDINGWINDOW:5:20 MINLEN:50". In order to generate a list of high-confidence SNP markers between Col and Ler, we adopted a strategy by combing the whole-genome alignment and short-read mapping[89,90]. First, the Col and Ler genomes were aligned to identify syntenic SNPs by SyRI[85]. Then, further checks and filters were applied to avoid the artificial and non-allelic SNPs by inGAP-family[33,39]. The sequencing reads of F1 and F2 samples were aligned to the TAIR10 Col reference genome by BWA[83]. The meiotic CO prediction and filtering of the poorly covered and potentially contaminated samples were performed using a sliding-window-based method[39,89]. Each sliding window was genotyped by the supporting reads of Col and Ler alleles. To avoid false genotyping, we selected 0.95 as the threshold allelic ratio for the determination of homozygosity in F1 hybrids. The final CO breakpoint was further refined by checking the genotype information of individual SNPs. Identified COs were manually checked at random using inGAP-family[39].

**SPO11-1-oligo, BrdU-seq, ChIP-seq, MNase-seq, DNase-seq, and ATAC-seq data analysis**. Short reads from public datasets (Supplementary Data 1) were quality-checked with FastQC. Specific 3′ adaptor and 5′ end sequences were trimmed before alignment by Cutadapt v1.9.1[91] as described[57]. For BrdU-seq and ATAC-seq datasets, the reads were processed with Trimmomatic to remove potential adaptor sequences and low-quality bases, with "LEADING:3 TRAILING:3 SLIDINGWINDOW:4:15 MINLEN:36". Duplicated reads were removed using BBMap (https://github.com/BioInfoTools/BBMap). Then, clean reads were aligned to the TAIR10 reference genome using Bowtie2 v2.2.8[92] with settings "–very-sensitive -k 10" for single-end datasets and further settings "–no-discordant –no-mixed" for paired-end datasets. The uniquely mapped reads were kept for subsequent analysis, which were processed by Samtools v1.9[93] and Sambamba v0.6.8[94]. For all sequencing data, coverage across the genome was evaluated and normalized with bins per million mapped reads (BPM) in bedGraph format using bamCoverage v3.4.3[95].

**Bisulfite sequencing data analysis**. The quality and adaptor sequencing of raw reads were examined by FastQC. The sequencing reads were mapped to the TAIR10 reference genome with Bismark v0.22.0[96], with the following setting: -q -bowtie2 -N 1 -L 24. Reads that mapped to multiple positions and duplicated alignments were removed. Methylated cytosines in the CG, CHG and CHH contexts and the level of methylation, were extracted for subsequent association analysis.

**Genome-wide CO distribution correlation analysis**. The chromosomal profiles of COs, genomic and epigenomic features were estimated in 50-kb windows along chromosomes. For a given window, the recombination frequency was normalized with the total CO number within the corresponding chromosome. Then, all of the COs, genomic and epigenomic data were smoothed with 40 nearby windows (total = 2 Mb) using the filter function (stats v3.6.2 package, with default parameter, a moving average strategy) and then normalized using the scale function (base v3.6.2 package) in the R environment. The non-linear correlation matrices were calculated using the nlcor package (https://github.com/ProcessMiner/nlcor)[97] in R, at the genome, chromosome-arm and pericentromeric scales, respectively. The constitution (peri-centromeres, centromeres and arms) of the TAIR10 reference genome was adopted from Underwood et al.[66]. Here, all the random forest models were trained using randomForest v4.6-14 package in R, with the setting of "mtry=3, importance=TRUE, na.action=na.omit, ntree=2000".

**Estimating nucleotide diversity and historical recombination rate**. For the sequence polymorphism data of 2029 Arabidopsis accessions from the 1001 Genomes Project[51] and the RegMap population[52], we first selected diallelic SNP positions with <20% missing data and >5% minimum allele frequency using VCFtools v0.1.16[98]. Then, we masked SNPs located in (i) tandem repeat regions (Tandem Repeats Finder output), (ii) repetitive elements and low-complexity regions (extracted from the masked TAIR10 reference genome), (iii) transposable elements (TAIR10 annotation) and (iv) centromeric regions (definition adopted from Underwood et al.[66]). Finally, we obtained a collection of 905,613 SNPs from 2029 accessions for CO frequency analysis. FastEPRR v2.0[99] was employed for estimating population recombination rates ($\rho = 4Ner$, where Ne is the effective population size and r is the recombination rate of the window), with 50-kb non-

overlapping window size. The nucleotide diversity of each 50-kb non-overlapping window along chromosomes was calculated using VCFtools. The geographical distribution of the 2029 Arabidopsis accessions was made by ggplot2 v3.3.5 package in R.

**Reporting summary**. Further information on research design is available in the Nature Research Reporting Summary linked to this article.

## Data availability

The raw sequencing data of individuals of Col, Ler inbred lines, and the Col/Ler F1 hybrid can be accessed in ArrayExpress under the accession numbers E-MTAB-11248, E-MTAB-11249, E-MTAB-11250, E-MTAB-11251, E-MTAB-11254, respectively. The public datasets used in this study are provided in the Supplementary Data 1. The list of COs identified in Col, Ler, F1 hybrid (female and male) and F2 hybrid can be found in Supplementary Data 2–5. The Col-0 TAIR10 reference genome is downloaded from the TAIR database [https://www.arabidopsis.org/]. The sequence polymorphism data of 2029 Arabidopsis accessions is downloaded from FigShare [https://figshare.com/projects/Imputation_of_3_million_SNPs_in_the_Arabidopsis_regional_mapping_population/72887]. Source data are provided with this paper.

## Code availability

The related code is available at GitHub [https://github.com/qclian/EMS_Col_Ler].

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

## Acknowledgements
We would like to thank the Max Planck Genome centre for DNA extraction, library preparation and sequencing, Hequan Sun and Wen-Biao Jiao for helpful discussions, Charles Underwood, Ian Henderson, and Andrew Tock for help with SPO11-1-oligo analysis and Wayne Crismani and Andrew Lloyd for critical reading of the manuscript. This work was support by core funding from the Max Planck Society and an Alexander von Humboldt Fellowship to Q.L.

## Author contributions
Q.L., K.S. and R.M. designed the research and analyzed the data. V.S. and B.W. generated plant materials. B.H. supervised the whole-genome sequencing work. S.D. performed and analyzed the cytology experiments. Q.L. and R.M. wrote the article with input from K.S.

## Funding

## Competing interests
The authors declare no competing interests.
