## [Peer Review File · Nature Communications]

The megabase-scale crossover landscape is largely independent of sequence divergenceReviewers' Comments:

Reviewer #1:

Remarks to the Author:

Crossover recombination is important not only for proper chromosome segregation but also for genetic diversity, and thus is tightly regulated at multiple levels. Lian et al. investigate the relationship between crossover distribution with polymorphisms and thus the regulation of crossovers at megabase-scale, and revealed that the megabase-scale recombination landscape is well correlated with chromatin features, mainly open chromatin, gene density, and CHH DNA methylation. Overall this study is well performed and interesting.

Concerns:

1. Crossover interference. Since there are lots of evidence suggest that the metric of crossover interference is the micron axis length (probably also including authors' previous publications), I am wondering what happens if authors plot CoC values against their axis lengths. And how is the CO landscape regulated by axis length (e.g. lines 151-152, and Figure 2)?
2. The megabase-scale recombination landscape is correlated with three features, open chromatin, gene density, and CHH DNA methylation. What is the relationship between these three features (and with other features), e.g. partially overlapped?

Reviewer #2:

Remarks to the Author:

This study presents an analysis of polymorphism density and its effect on meiotic crossover landscapes along the Arabidopsis thaliana chromosomes. A low density of polymorphic markers are introduced into the Col and Ler parents using EMS. Markers are in the range of 471 to 955 per genome. Reciprocal backcrossing performed to look at male and female meiosis separately, using between 251 and 309 progeny per population. In total 3,155 and 2,004 crossovers mapped in the Col and Ler populations. They also generate a control Col x Ler F2 population, and compare to other published datasets. Overall the data looks good, is well performed and the analysis is appropriate. This also represents a novel experimental design that will be of interest to the field.

However, I do take issue with the emphasis placed in the paper and title that polymorphism has 'no effect'. I dont think this is completely true, although I can appreciate the point the authors are trying to make. I think a more nuanced stance should be taken throughout.

Fig 1 D, E and F. There are some differences in the profiles that I feel should be more specifically commented on in the results section. First the distal bias in the Ler population looks different to Col. For example the left arm of chromosome 1 is barely higher in Ler male compared to female, whereas the right arm is very elevated – I wonder why this would be? It seems generally true for the other chromosomes also.

Also generally, Im not convinced the landscapes are 'the same' in the three cases. The broad trend is the same along the chromosome arms, but the topology looks quite different! For example, compare the right arm of chr5 in males across the three populations – the topology looks different to me. Perhaps the authors can provide some form of statistical test that these profiles are indistinguishable or different? Another case would be the left telomeric end of chr4, which has a very high male recombination rate, that seems higher than either the Col or Ler population. Could these differences not be ascribed to polymorphism? I would agree that at the broad scale there is a significant elevation in the pericentromere region in all populations, and I agree that this would appear not to be driven by polymorphism, although it correlates with it.

The authors make the point that know that rearrangements including inversions and translocations have large suppressive effects on crossovers at the megabase scale. This would seem to me to argue against the title of the paper, and the key message being conveyed. Actually divergence does have an effect at the megabase scale as inversions etc show.

I think in Figure 2B there are again differences evident; the right + left arms of chr5 are higher in Col for example, and this is true on many of the other chrs also.

In Fig 2C – please add values to the y axis – is it the same in each case? ‘Normalized value’ needs to be more descriptive, as the plots seem to be showing different things – eg SNPs in one case and crossovers in the others. I have the same comment as before that the crossover profiles do not look identical to me. I think again applying some kind of confidence test is important – are these profiles actually different or not? For the SNP profile, it might also be interesting to plot the density of the Col and Ler derived EMS markers?

Figure 4. The positive effect of CHH is probably the most novel observation here. What do the authors think about that finding?

Another important point that should be considered is that the vast majority of datasets compared with are generated from a Col accession. We dont know how different chromatin, or these other features, are in Ler. I think its OK to use these datasets, but a caveat should be added to the text somewhere noting this.

All populations showed heterochiasmy. I note that the authors observe: “However, an even lower level of COs is observed in the hybrid (2.79, $p = 0.0002$), suggesting that an additional phenomenon is responsible for the reduced CO frequency specifically in female hybrids.” Could this reflect an inhibitory effect of a higher polymorphism level specifically in female meiosis? This point is raised in the discussion (lines 286-292) where the authors also acknowledge that polymorphism is affecting recombination potentially more in the female side. I think they are right, but doesn’t this rather detract from the conclusion that divergence has ‘no effect’? The authors even propose a mechanisms for this effect due to Class II event frequency in the female (I agree this is the likely explanation).

Line 155-156 – please provide references for the ‘previous findings’ mentioned.

Line 95 – please employ a better description that ‘pretty uniformly’ – its not clear what this means.

Line 24 – ‘absence of polymorphisms’ this is not correct – there are a few hundred markers, so its low but not absent – please correct. This should also be corrected on lines 171, 188 and all other instances.

Lines 228-230. Its nice to see these relationships, although I note that most of these conclusions have already been reached by multiple other prior publications.

Line 277 – could be worth citing the recent Nature paper from the Weigel group on this topic? similar conclusions have been reached in other publications aswell (eg Harberd lab in Genome Res).

Line 283 -284. In addition to HEI10, the Ziolkowski group recently reported that a second Col/Ler recombination QTL exists in SNI1 (PNAS paper). Please add a citation for this. In Figure 1 its interesting to see that Ler has consistently fewer crossovers than Col in relation to this.

Line 432 – please change to ‘SPO11-1-oligos analysis’.

Reviewer #3:

Remarks to the Author:

Lian et al. present a smart approach for the genome-wide study of meiotic recombination in homozygous inbred lines, based on the introduction of a small number of markers. In general, investigating the underlying causes of variation in meiotic recombination in plants is a topic with both fundamental importance for genetics, and practical relevance for plant breeding. Specifically, this paper focuses on the role of sequence divergence; mixed findings have been reported in literature on this role, as clearly described in the introduction of the manuscript. As such, findings in this paper contribute to addressing a relevant and important issue. Based on a comparison of the recombination landscape in two different *Arabidopsis thaliana* inbred lines with the landscape in a hybrid and with historical recombination rate patterns in *Arabidopsis*, they conclude that polymorphism density is not a major determinant of the crossover landscape. In addition, they investigate how a couple of simple features (chromatin accessibility, gene density, and DNA methylation) are sufficient to explain a large part of the megabase-scale recombination landscape in *Arabidopsis*.

One limitation of the approach presented here is that it can only be used to analyze the large scale recombination landscape. Although "large-scale" and "megabase scale" are clearly mentioned in the manuscript, it would still be useful if some more thought would be given to how large-scale and fine-scale influence of sequence polymorphisms are or are not related. That would then allow somewhat more elaborate discussion of previous findings at a smaller scale; one example would e.g. the results on the role of MSH2 in regions of higher sequence diversity in Blackwell 2020; this paper is referenced (ref 21) but findings in this and similar papers on the local role of sequence divergence are not taken into account in the discussion.

One other question I have related to the chosen approach is whether bias in terms of where the EMS mutations are introduced could influence the results (as a kind of confounding factor). Although some results are reported suggesting that mutation position is uniform, I did not see a clear analysis on whether mutation position somehow was correlated with variation in sequence divergence. This should be analyzed (one could e.g. consider the sequence divergence between Col and Ler in genome regions where the EMS mutations occur compared to other regions). If there is such correlation, it should be discussed how this potentially influences the reported results.

Related to the data analysis, I see a few major issues that need to be addressed:

- If I understand correctly, the input for random forest are the CO distributions obtained as described in methods in "Genome-wide CO distribution correlation analysis". There, 50kb windows are described, and data is smoothed with 40 nearby windows. This means that individual windows are strongly related to each other, which will lead to strongly overestimated prediction performance of the random forest model. Although training and test set are being used, these are apparently randomly sampled, which will mean that for a given interval in the test set, there will be several very related intervals in the training. To deal with this, one could should devise a training-test set (or cross-validation approach) which takes the relatedness between the intervals into account. One simple strategy could be to use a leave-one-chromosome out cross-validation strategy: use data from four chromosomes for training, the remaining chromosome for testing, and repeat this five times, once for each chromosome. More involved strategies would also be possible, as long as one makes sure that from a given chromosome, the intervals that are in the test set are not related to the intervals in the training set (i.e. not close together on the chromosome).

- A rather non-standard "non-linear correlation" is used. A reference is given to a github for this method, which shows that locally linear regression is used. It would be good to have a more regular reference to scientific literature, and, more importantly, some explanation as to why this approach was chosen, and how results would potentially be different if linear correlation is used.

- The window size used in the analysis clearly needs to be large here given the input data; however, how is the exact value chosen and what happens when it is made smaller?

Some additional minor issues:

- Methods p15/16 mentions "filter" function for smoothing – please describe what this does.
- In Fig 3D-F the correlation analysis is performed separately for euchromatin vs heterochromatin, which is indeed appropriate. However it is not clear if the random forest based prediction could also be influenced by euchromatin vs heterochromatin state; this should be analyzed.

Reviewer #4:

Remarks to the Author:

The manuscript by Lian et al. addresses the question of the effect of DNA sequence diversity on CO rates using the Arabidopsis model. There are several interesting aspects of this work, for example the comparison of male and female recombination rates, but a number of issues require correction and refinement.

The fact that DNA sequence polymorphism does not change the overall shape of recombination landscape has been known for a while. There are detailed studies in tomato and maize that relied on analysis of recombination nodules (Anderson et al., 2004, *Genetics* 166: 1923–1933; Anderson et al., 2003, *Genetics* 165: 849–865; Koo et al, 2008; *Genetics* 179:1211–1220; Sherman & Stack, 1995, *Genetics* 141: 673–708 and some others). These studies contain CO data with very high resolution and show essentially the same results as Lian et al. None of them, however, are cited in the current manuscript.

The authors discuss at length the discrepancy in CO rates between male and female recombination in the Ler x Col hybrid and propose that the ratio of Class I to Class II COs may be changed. This can be easily tested with immunolocalization. Immunolocalization in male meiocytes is routine and there is now a robust protocol for immunolocalization in female meiocytes (Escobar-Guzman et al., 2015, *Nature Protocols* 10:1535–1542). Although immunolocalizing Class II COs is difficult, there is a good anti-MLH1 antibody for Arabidopsis.

The authors conclude that 85% of the megabase-scale recombination landscape can be explained by just three factors, gene density, chromatin accessibility and DNA methylation. What would be the remaining 15%? I suspect gene density affects recombination landscape by the virtue of promoters displaying open chromatin configuration, which should correlate with chromatin features. Can gene density in authors algorithm be supplanted by other chromatin features?

More information on the resolution of CO mapping in the hybrid is needed. In what fraction of COs is the resolution sufficient to assign them to promoters? What fraction of COs were used to generate Figure 7D? Was their distribution uniform along chromosomes?

Is chromatin accessibility measured in whole flowers (line 197) indeed able to explain CO distribution? Meiocytes must constitute a small fraction of whole-flower tissue.

How were the low-sequence-coverage markers imputed and corrected (line 354/355)?

What was the rationale for choosing 50kb as the window size for genome-wide comparisons (line 399)? 50kb is quite a bit given the gene density in Arabidopsis. The authors could try the analysis with the fraction of COs that can be mapped more precisely and then use window sizes smaller than 50kb.

Reviewer #1 (Remarks to the Author):

Crossover recombination is important not only for proper chromosome segregation but also for genetic diversity, and thus is tightly regulated at multiple levels. Lian et al. investigate the relationship between crossover distribution with polymorphisms and thus the regulation of crossovers at megabase-scale, and revealed that the megabase-scale recombination landscape is well correlated with chromatin features, mainly open chromatin, gene density, and CHH DNA methylation. Overall this study is well performed and interesting.

> We are pleased that reviewer#1 appreciates the study.

Concerns:

1. Crossover interference. Since there are lots of evidence suggest that the metric of crossover interference is the micron axis length (probably also including authors' previous publications), I am wondering what happens if authors plot CoC values against their axis lengths. And how is the CO landscape regulated by axis length (e.g. lines 151-152, and Figure 2)?

> It is indeed very clear that the mechanistic metric of crossover interference is micron axis length, we fully agree on that point. Unfortunately, we cannot place the genetically detected crossovers along the chromosome axis, preventing us from directly plotting CoC curves on a μm scale. However, assuming that the $\mu\text{M}/\text{Mb}$ is homogeneous along chromosomes, we can convert our CoC curves into the μm space, which is different in males and females (ratio=1.70 in the Arabidopsis Col/Ler hybrid; Drouaud et al, PLoS Genetics 2007). After this transformation, the male and female CoC curves are very similar, confirming that the observed difference in crossover interference between males and females is due to the difference in chromosome axis length. We have included this analysis as Supplemental Figure S12 and have added an explanation of this important point in the main text (lines 177–182).

2. The megabase-scale recombination landscape is correlated with three features, open chromatin, gene density, and CHH DNA methylation. What is the relationship between these three features (and with other features), e.g. partially overlapped?

> Yes, these three features are correlated with each other and with other features (Figure 4A). For example, open chromatin (measured with ATAC-seq) and gene density have a non-linear correlation coefficient of 0.74. However, they only partially overlap as, for example, open chromatin and gene density explain much more of the variation than open chromatin or gene density alone (Figure 4C). As stated in the manuscript, we do not think that this is indicative of direct causality; we only claim that these features are best linked to the actual chromatin states that favour crossover formation. One illustration of this is the observation that if one of the major features is excluded from the analysis (e.g., gene density), other features can then account for the variation of crossover distribution (Supplemental Figure S22).

Reviewer #2 (Remarks to the Author):

This study presents an analysis of polymorphism density and its effect on meiotic crossover landscapes along the Arabidopsis thaliana chromosomes. A low density of polymorphic markers are introduced into the Col and Ler parents using EMS. Markers are in the range of 471 to 955 per genome. Reciprocal backcrossing performed to look at male and female meiosis separately, using between 251 and 309 progeny per population. In total 3,155 and 2,004 crossovers mapped in the Col and Ler populations. They also generate a control Col x Ler F2 population, and compare to other published datasets. Overall the data looks good, is well performed and the analysis is appropriate. This also represents a novel experimental design that will be of interest to the field.

> We are pleased that reviewer#2 recognizes the originality of the experimental design and the quality of the data and the analysis.

However, I do take issue with the emphasis placed in the paper and title that polymorphism has 'no effect'. I don't think this is completely true, although I can appreciate the point the authors are trying to make. I think a more nuanced stance should be taken throughout.

> In our willingness to deliver a clear message, we probably oversimplified at the cost of lack of accuracy. Thank you for pointing this out. We thus modified the title to “The megabase-scale crossover landscape is largely independent of sequence divergence”, which we believe is more appropriate. We also modified the main text with the aim of eliminating all instances where we potentially overstate our findings (see also below).

Fig 1 D, E and F. There are some differences in the profiles that I feel should be more specifically commented on in the results section. First the distal bias in the Ler population looks different to Col. For example the left arm of chromosome 1 is barely higher in Ler male compared to female, whereas the right arm is very elevated – I wonder why this would be? It seems generally true for the other chromosomes also.

> Indeed, the female/male difference is less pronounced in Ler, notably in distal regions. This may be due to a generally lower frequency of COs in Ler compared to Col and because trans-factors (e.g., HEI10) tend to affect more distal regions (Ziolkowski et al, 2017). It should also be noted that the Ler profile tends to have a larger interval of confidence, notably at telomeres, because of a slightly smaller sample size and marker set than the two other genotypes. We have added these considerations to the revised manuscript (lines 156-161).

Also generally, I'm not convinced the landscapes are 'the same' in the three cases. The broad trend is the same along the chromosome arms, but the topology looks quite different! For example, compare the right arm of chr5 in males across the three populations – the topology looks different to me. Perhaps the authors can provide some form of statistical test that these profiles are indistinguishable or different? Another case would be the left telomeric end of chr4, which has a very high male recombination rate, that seems higher than either the Col or Ler population. Could these differences not be ascribed to polymorphism? I would agree that at the broad scale there is a significant elevation in the pericentromere region in all populations, and I agree that this would appear not to be driven by polymorphism, although it correlates with it.

> It is indeed a simplification to say that the profiles are the same. We revised the manuscript to ensure that we do not overstate our findings. We added an analysis based on linear correlations to show that the profiles are very similar (Figure 3D–E). Strikingly, the correlation between the recombination landscapes of Col and Hybrid, or *Ler* and Hybrid, is higher than the recombination landscapes of the two pure lines. Another striking observation is the inversion that clearly stands out in this analysis (Figure 3D). We also now comment that while they are similar, they are not identical, leaving open the possibility that sequence divergence has some effect on the megabase-scale crossover landscapes.

The authors make the point that know that rearrangements including inversions and translocations have large suppressive effects on crossovers at the megabase scale. This would seem to me to argue against the title of the paper, and the key message being conveyed. Actually divergence does have an effect at the megabase scale as inversions etc show.

> We modified the title to “The megabase-scale crossover landscape is largely independent of sequence divergence”, which we think is fair, notably because inversions account for only a very limited part of the genome (Jiao et al, Nat Common 2020). While the effect of inversions is not specifically addressed in the title, this is very clearly stated in the abstract.

I think in Figure 2B there are again differences evident; the right + left arms of chr5 are higher in Col for example, and this is true on many of the other chrs also.

> We recognize that the frequency of crossovers is higher in Col, probably due to natural variation in trans factors such as HEI10. To analyse the profiles independently of the global rates, we normalized the profiles (Figure 3C) (see below). We also recognize that despite the striking similarities, there are also differences in the profiles, reflected by the non-complete correlations. This imperfect correlation could be due to a genuine biological effect or to unavoidable imprecision in the estimation of recombination rates. We revised the manuscript to be more cautious.

In Fig 2C – please add values to the y axis – is it the same in each case? ‘Normalized value’ needs to be more descriptive, as the plots seem to be showing different things – eg SNPs in one case and crossovers in the others. I have the same comment as before that the crossover profiles do not look identical to me. I think again applying some kind of confidence test is important – are these profiles actually different or not? For the SNP profile, it might also be interesting to plot the density of the Col and ler derived EMS markers?

> We modified the figure legend to make the analysis clearer: The relative crossover frequency (the number of crossovers in the given window divided by the total crossover number within the respective chromosome) was calculated for each individual chromosome. Then, the crossover and SNP profiles were all z-score-normalized using the scale function in the R environment. In addition, we added Spearman’s correlation values of the CO landscape of the pure lines in Figure 3D–E, which reveal a strong correlation between the landscapes. Concerning SNP density, we added the plot suggested in Supplemental Figure S2E. We also showed that there is a very low correlation between marker (i.e., mutation) density and COs (Supplemental Figure S16; see also responses to reviewer #3).

Figure 4. The positive effect of CHH is probably the most novel observation here. What do the authors think about that finding?

> A positive correlation between crossovers and DNA methylation patterns (including CHH methylation) in chromosome arms has already been reported by Choi et al, 2018, Genome Res as well as by Lambing et al, 2020, Plant Cell. In the updated manuscript we are now explicitly referring to these earlier works.

Another important point that should be considered is that the vast majority of datasets compared with are generated from a Col accession. We don't know how different chromatin, or these other features, are in Ler. I think it's OK to use these datasets, but a caveat should be added to the text somewhere noting this.

> We agree. Accordingly, in Figure 4, we compared the CO landscape and genomic features exclusively in the Col accession, in which all the data were produced. We modified the text to make this point clearer (lines 209-211).

All populations showed heterochiasmy. I note that the authors observe: "However, an even lower level of COs is observed in the hybrid (2.79, $p = 0.0002$), suggesting that an additional phenomenon is responsible for the reduced CO frequency specifically in female hybrids." Could this reflect an inhibitory effect of a higher polymorphism level specifically in female meiosis? This point is raised in the discussion (lines 286-292) where the authors also acknowledge that polymorphism is affecting recombination potentially more in the female side. I think they are right, but doesn't this rather detract from the conclusion that divergence has 'no effect'? The authors even propose a mechanism for this effect due to Class II event frequency in the female (I agree this is the likely explanation).

> We are pleased that the reviewer agrees with our interpretations. Again, the title was a bit too simple and was modified accordingly.

Line 155-156 – please provide references for the 'previous findings' mentioned.

> We added the relevant references.

Line 95 – please employ a better description than 'pretty uniformly' – it's not clear what this means.

> This wording was indeed not very accurate. We modified to "randomly" and added an analysis that showed that markers are indeed randomly distributed along chromosomes (Supplemental Figure S2).

Line 24 – 'absence of polymorphisms' this is not correct – there are a few hundred markers, so it's low but not absent – please correct. This should also be corrected on lines 171, 188 and all other instances.

> We agree that the "absence of polymorphisms" is formally not correct, even if the introduced polymorphisms represent <0.1% of the differences between Col and Ler. We thus modified the text to "quasi-absence of polymorphism".

Lines 228-230. It's nice to see these relationships, although I note that most of these conclusions have already been reached by multiple other prior publications.

> We agree and make this clear in the revised manuscript with relevant references. The result that was more surprising to us is that such a large part of the variation can be accounted for by so few features.

Line 277 – could be worth citing the recent Nature paper from the Weigel group on this topic? Similar conclusions have been reached in other publications as well (eg Harberd lab in Genome Res).

> Yes, definitely. The Weigel work was published just after we submitted this manuscript and is now cited in the revised manuscript together with the Harberd work.

Line 283 -284. In addition to HEI10, the Ziolkowski group recently reported that a second Col/Ler recombination QTL exists in SNII (PNAS paper). Please add a citation for this. In Figure 1 its interesting to see that Ler has consistently fewer crossovers than Col in relation to this.

> We have included this work in the revised manuscript.

Line 432 – please change to ‘SPO11-1-oligos analysis’.

> Done.

Reviewer #3 (Remarks to the Author):

Lian et al. present a smart approach for the genome-wide study of meiotic recombination in homozygous inbred lines, based on the introduction of a small number of markers. In general, investigating the underlying causes of variation in meiotic recombination in plants is a topic with both fundamental importance for genetics, and practical relevance for plant breeding. Specifically, this paper focuses on the role of sequence divergence; mixed findings have been reported in literature on this role, as clearly described in the introduction of the manuscript. As such, findings in this paper contribute to addressing a relevant and important issue. Based on a comparison of the recombination landscape in two different Arabidopsis thaliana inbred lines with the landscape in a hybrid and with historical recombination rate patterns in Arabidopsis, they conclude that polymorphism density is not a major determinant of the crossover landscape. In addition, they investigate how a couple of simple features (chromatin accessibility, gene density, and DNA methylation) are sufficient to explain a large part of the megabase-scale recombination landscape in Arabidopsis.

> Thank you for pointing out the importance of our work.

One limitation of the approach presented here is that it can only be used to analyze the large scale recombination landscape. Although “large-scale” and “megabase scale” are clearly mentioned in the manuscript, it would still be useful if some more thought would be given to how large-scale and fine-scale influence of sequence polymorphisms are or are not related. That would then allow somewhat more elaborate discussion of previous findings at a smaller scale; one example would e.g. the results on the role of MSH2 in regions of higher sequence diversity in Blackwell 2020; this paper is referenced (ref 21) but findings in this and similar papers on the local role of sequence divergence are not taken into account in the discussion.

> We agree that local polymorphism density and mismatch recognition play important roles in the local placement of COs as shown by Blackwell et al. In turn, a local effect does not necessarily influence the global (megabase-scale) distribution. While some chromosome-scale redistribution could be observed in the *msh2* (mismatch recognition) mutants, the overall recombination landscapes between the mutant and the wild type remained highly correlated. In the revised version of the manuscript, we now discuss these points in more detail.

One other question I have related to the chosen approach is whether bias in terms of where the EMS mutations are introduced could influence the results (as a kind of confounding factor). Although some results are reported suggesting that mutation position is uniform, I did not see a clear analysis on whether mutation position somehow was correlated with variation in sequence divergence. This should be analyzed (one could e.g. consider the sequence divergence between Col and Ler in genome regions where the EMS mutations occur compared to other regions). If there is such correlation, it should be discussed how this potentially influences the reported results.

> We included as Supplemental Figure S2 a plot showing the distributions of the markers, SNPs and crossovers, and we added a correlation analysis between the crossover landscape and polymorphisms in Supplemental Figure S16, which showed that there is no significant correlation between them.

Related to the data analysis, I see a few major issues that need to be addressed:

- If I understand correctly, the input for random forest are the CO distributions obtained as described in methods in “Genome-wide CO distribution correlation analysis”. There, 50kb windows are described,

and data is smoothed with 40 nearby windows. This means that individual windows are strongly related to each other, which will lead to strongly overestimated prediction performance of the random forest model. Although training and test set are being used, these are apparently randomly sampled, which will mean that for a given interval in the test set, there will be several very related intervals in the training. To deal with this, one could should devise a training-test set (or cross-validation approach) which takes the relatedness between the intervals into account. One simple strategy could be to use a leave-one-chromosome out cross-validation strategy: use data from four chromosomes for training, the remaining chromosome for testing, and repeat this five times, once for each chromosome. More involved strategies would also be possible, as long as one makes sure that from a given chromosome, the intervals that are in the test set are not related to the intervals in the training set (i.e. not close together on the chromosome).

> We indeed overlooked this flaw in the initial analysis, thank you for pointing this out. We now analyse the power of the random forest prediction by using four chromosomes as the training set, to predict the landscape of the fifth chromosome. This analysis was done separately for all five chromosomes and is shown in Figure 4F. The prediction recapitulates well the landscapes with a global linear correlation of 0.6.

- A rather non-standard “non-linear correlation” is used. A reference is given to a github for this method, which shows that locally linear regression is used. It would be good to have a more regular reference to scientific literature, and, more importantly, some explanation as to why this approach was chosen, and how results would potentially be different if linear correlation is used.

> To compare the crossover landscapes between inbreds and the hybrid, we use linear correlation (Spearman’s, Figure 3D-E) to determine the similarity of the patterns. However, we do not expect the relationship between crossover landscape and genomic/epigenomic features to always be linear. For example, increasing the value of one of these features may favour COs up until a certain point and then decrease COs with higher values (an inverted U shape). To capture such effects, we used a non-linear correlation (Figures 3F and 4A). To address any putative differences between linear and non-linear correlation values, we also now included linear correlation analyses of the same data (Supplemental Figure S14 and S16).

- The window size used in the analysis clearly needs to be large here given the input data; however, how is the exact value chosen and what happens when it is made smaller?

> Following the resolution of COs detected in Col and Ler (Supplemental Figure S6), we selected 2 Mb as window size as this covers the majority of the COs (98.8% in Col and 90.2% in Ler). Choosing a smaller window size would have introduced artifacts, as the positions of many COs would be known with intervals of confidence that would be larger than the window size.

Some additional minor issues:

- Methods p15/16 mentions “filter” function for smoothing – please describe what this does.

The filter function (stats package in R), with the default parameter, is used to smooth the dataset by a moving average. This is specified in the revised manuscript.

- In Fig 3D-F the correlation analysis is performed separately for euchromatin vs heterochromatin, which is indeed appropriate. However it is not clear if the random forest based prediction could also be influenced by euchromatin vs heterochromatin state; this should be analyzed.

>The random forest analysis was now performed either on chromosome arms only (Figure 4B–C, Supplemental Figure S21) or genome-wide (Figure 4D–F).

Reviewer #4 (Remarks to the Author):

The manuscript by Lian et al. addresses the question of the effect of DNA sequence diversity on CO rates using the Arabidopsis model. There are several interesting aspects of this work, for example the comparison of male and female recombination rates, but a number of issues require correction and refinement.

The fact that DNA sequence polymorphism does not change the overall shape of recombination landscape has been known for a while. There are detailed studies in tomato and maize that relied on analysis of recombination nodules (Anderson et al., 2004, Genetics 166: 1923–1933; Anderson et al., 2003, Genetics 165: 849–865; Koo et al, 2008; Genetics 179:1211–1220; Sherman & Stack, 1995, Genetics 141: 673–708 and some others). These studies contain CO data with very high resolution and show essentially the same results as Lian et al. None of them, however, are cited in the current manuscript.

> We were of course aware of these beautiful studies and are happy to cite them in the revised manuscript. However, we respectfully disagree that these approaches offer a powerful way to compare recombination in hybrids and inbreds. Comparisons of recombination nodule distribution with genetic data are constrained by the comparison of DNA (genetics) space with μm space (synaptonemal complex). FISH can partially address this limitation, but with limited resolution (“The calculated centimorgan: micrometer ratio trends are similar to the RN distribution” Koo et al, 2008). These data showed that the global pattern is similar, and this result is included in the revised manuscript. Generally, comparing data obtained with very different approaches is challenging, as exemplified by the mismatches between recombination nodule maps and genetic maps discussed in Anderson et al., 2003 and Chang et al, 2007. Our approach offers a solution to these problems by directly comparing genetic data in the three genotypes. This allows a finer comparison of the numbers and profiles and, in addition, facilitates analysis of both male and female recombination.

The authors discuss at length the discrepancy in CO rates between male and female recombination in the Ler x Col hybrid and propose that the ratio of Class I to Class II COs may be changed. This can be easily tested with immunolocalization. Immunolocalization in male meiocytes is routine and there is now a robust protocol for immunolocalization in female meiocytes (Escobar-Guzman et al., 2015, Nature Protocols 10:1535–1542). Although immunolocalizing Class II COs is difficult, there is a good anti-MLH1 antibody for Arabidopsis.

> This is a great suggestion. Because of the scarcity of female meiocytes versus male meiocytes, collecting multiple images of female meiocytes at the right stage is laborious, but we have added the analysis of MLH1 foci in both females and males of Col, Ler and hybrids. The results are consistent with our genetic data, confirming the heterochiasmy in Col, Ler and hybrids. It also confirmed that in male meiosis, there are more COs in Col than in Ler and that the hybrids have intermediate levels of COs. In female meiosis, we did not detect differences between the genotypes, but this is probably because this analysis is not as powerful as our genetic analysis: The expected differences are small (2.8 COs in female hybrids compared to 3.34 in female Ler) but could be captured with our approach because we could type a large number of meiotic events (n=428, and n=253). It is, unfortunately, unrealistic to analyse such a large number of female meiocytes with cytology. This analysis is included in the revised manuscript (Supplemental Figure S9).

The authors conclude that 85% of the megabase-scale recombination landscape can be explained by just three factors, gene density, chromatin accessibility and DNA methylation. What would be the remaining

15%? I suspect gene density affects recombination landscape by the virtue of promoters displaying open chromatin configuration, which should correlate with chromatin features. Can gene density in authors algorithm be supplanted by other chromatin features?

> By taking into account more chromatin features, more variation can be explained (~95%, Figures 4C and 4E). Yes, gene density can be supplanted by other features, as shown in Supplemental Figure S22. In fact, many of the features that collectively predict the CO landscapes are also correlated with each other. We do not claim that any of these features are causal and directly determine the CO landscape, but rather that these features capture the chromatin state that shapes the landscape.

More information on the resolution of CO mapping in the hybrid is needed. In what fraction of COs is the resolution sufficient to assign them to promoters? What fraction of COs were used to generate Figure 7D? Was their distribution uniform along chromosomes.

> The resolution of the CO intervals detected in the female and male hybrid (Supplemental Figure S6A) is 2.4 kb and 3.8 kb on average. We selected COs with interval size ≤ 3 kb for the fine-scale analysis, including 82.2% and 72.8% of the COs in female and male hybrids, respectively. We added the chromosomal distribution of high-resolution COs in female and male hybrids in Supplemental Figure S7D. The distribution of high-resolution COs is similar to the distribution of the entire dataset.

Is chromatin accessibility measured in whole flowers (line 197) indeed able to explain CO distribution? Meiocytes must constitute a small fraction of whole-flower tissue.

> We collected from the literature the ATAC-seq data from seedlings, leaves, vegetative cells, and microspores, and analysed these data in the same way as the ATAC-seq data (from flowers) used in our study. As shown in Supplemental Figure S23, the open chromatin profiles of the different tissues are highly correlated with each other at the megabase scale. This suggests that the mega-base chromatin landscape is stable throughout development and that this stable landscape is a major driver of the CO landscape.

How were the low-sequence-coverage markers imputed and corrected (line 354/355)?

> We adopted a hierarchical clustering-based sliding window method for phasing mutation markers, with a window size of 10 markers and a step size of 1 marker. Each marker was involved in several (1–10) windows for clustering and phasing. Then, a voting strategy that integrated the result of multiple clustering (support window count table, Supplemental Figure S1) was performed to evaluate the confidence of the genotype of the marker. For the first and last 5–9th markers, a support rate of 0.9 was used to impute and correct the genotype of the marker if it was not covered or only poorly covered by sequencing reads. For all other non-covered or poorly covered markers, a support rate of 0.8 was used. We clarified this in the revised Methods section.

What was the rationale for choosing 50kb as the window size for genome-wide comparisons (line 399)? 50kb is quite a bit given the gene density in Arabidopsis. The authors could try the analysis with the fraction of COs that can be mapped more precisely and then use window sizes smaller than 50kb.

> Given the CO resolution in Col and Ler, which by construction is low (limited number of markers), we used a 2 Mb window size, and a 50 kb step size (see also responses to reviewer #3). We modified the Methods section to make this clearer.

Reviewers' Comments:

Reviewer #1:

Remarks to the Author:

The authors have addressed my concerns. It is a beautiful work.

Reviewer #2:

None

Reviewer #3:

Remarks to the Author:

My previous concerns have all been addressed.

Reviewer #4:

Remarks to the Author:

I do appreciate the authors attention to revising the manuscript and I believe the revisions have made it better. I think it is very appropriate that the authors mention previous work on recombination nodules in the revised manuscript version. However, I do not think it is appropriate to downplay the validity of the previously published results. In particular, the comment about low resolution of recombination nodule mapping is incorrect, as they were examined with high precision using electron microscopy. In fact, the resolution in the 2003 Anderson et al. work in terms of chromosome segments was very similar to what Lien et al. present (165 bins on average per chromosome vs 100 – 200 bins on average per chromosome in Lien et al.). Similarly, the statement of limited number of individual that were analyzed in the previous work is incorrect. The author need to revise these statements.

We are very pleased with the positive evaluation of our revised manuscript.

Following reviewer #4 suggestion, we have deleted the negative statement on recombination nodule analysis. We have also edited the manuscript to follow the formatting instructions.

Thank you for your interest in our work, and we are looking forward to celebrating the final acceptance of the manuscript.